# *In operando* NMR investigations of the aqueous electrolyte chemistry during electrolytic $CO_2$ reduction

Sven Jovanovic [1✉], Peter Jakes[1], Steffen Merz[1], Davis Thomas Daniel[1], Rüdiger-A. Eichel [1,2] & Josef Granwehr [1,3]

The electrolytic reduction of $CO_2$ in aqueous media promises a pathway for the utilization of the green house gas by converting it to base chemicals or building blocks thereof. However, the technology is currently not economically feasible, where one reason lies in insufficient reaction rates and selectivities. Current research of $CO_2$ electrolysis is becoming aware of the importance of the local environment and reactions at the electrodes and their proximity, which can be only assessed under true catalytic conditions, i.e. by *in operando* techniques. In this work, multinuclear *in operando* NMR techniques were applied in order to investigate the evolution of the electrolyte chemistry during $CO_2$ electrolysis. The $CO_2$ electroreduction was performed in aqueous $NaHCO_3$ or $KHCO_3$ electrolytes at silver electrodes. Based on [13]C and [23]Na NMR studies at different magnetic fields, it was found that the dynamic equilibrium of the electrolyte salt in solution, existing as ion pairs and free ions, decelerates with increasingly negative potential. In turn, this equilibrium affects the resupply rate of $CO_2$ to the electrolysis reaction from the electrolyte. Substantiated by relaxation measurements, a mechanism was proposed where stable ion pairs in solution catalyze the bicarbonate dehydration reaction, which may provide a new pathway for improving educt resupply during $CO_2$ electrolysis.

[1] Institute of Energy and Climate Research - Fundamental Electrochemistry (IEK-9), Forschungszentrum Jülich GmbH, Willhelm-Johnen-Straße, Jülich, Germany. [2] Institute of Physical Chemistry (IPC), RWTH Aachen University, Aachen, Germany. [3] Institute of Technical and Macromolecular Chemistry (ITMC), RWTH Aachen University, Aachen, Germany. ✉email: s.jovanovic@fz-juelich.de

At the time of its discovery in the late 19th century, $CO_2$ electrolysis was considered a niche method to resemble the biological incorporation of carbon dioxide in plants[1]. However, against the backdrop of rapidly increasing carbon dioxide levels in the atmosphere, the electrochemical conversion of $CO_2$ has assumed a new role of urgency to address the looming problems of anthropogenic climate change[2–4]. Thus, during the last decades, huge efforts have been made towards optimizing $CO_2$ electrolysis[5–7]. One finding was that $CO_2$ electrolysis can selectively yield products depending on the metal electrocatalyst[8]. At gold and silver electrodes, for instance, carbon monoxide (CO) is formed[5,9]. Furthermore, the product selectivity can be optimized through the electrolyte and by varying experimental conditions[7,10,11]. It is also established that the Hydrogen Evolution Reaction (HER) is a generally undesired process in aqueous electrolytes, which competes with the $CO_2$ reduction reaction[12,13].

Even though CO, which is one possible product of $CO_2$ electrolysis, lacks the direct usability of, e.g., short chain alcohols, it is a highly versatile educt in the chemical industry. In the Fisher–Tropsch process, the mixture of CO and $H_2$, also referred to as syngas, can be used for the synthesis a wide array of organic product molecules[14]. Moreover, CO can be easily separated from the aqueous electrolyte due to its low solubility in water. Among all catalysts yielding CO, silver is the prime choice because of its moderate cost, high selectivity and availability[5,7,8,15].

The relevant thermodynamic potentials $\phi_0^\ominus$ *vs.* Normal Hydrogen Electrode (NHE) of the $CO_2$ electrolysis at pH 7, 25 °C, 1 atm gas pressure and 1 mol $L^{-1}$ bicarbonate electrolyte are[2,16]

$$CO_2 + 2H^+ + 2e^- \longrightarrow CO + H_2O \quad \phi_0^\ominus = -0.53V \quad (1)$$

$$CO_2 + e^- \longrightarrow CO_2^{\bullet-} \quad \phi_0^\ominus = -1.90V \quad (2)$$

$$2H^+ + 2e^- \longrightarrow H_2 \quad \phi_0^\ominus = -0.41V \quad (3)$$

Since the equilibrium potentials of $CO_2$ reduction to CO (Eq. (1)) and the HER (Eq. (3)) are within the same range, the two reactions are expected to compete with one another. However, the electrolytic $CO_2$ reduction undergoes an intermediate step with a high activation barrier, whereby a $CO_2^{\bullet-}$ radical is formed (Eq. (2)). Therefore, the energy efficiency of the $CO_2$ electrolysis, defined as the ratio of the difference in free energy between products and educts and the energy consumed in the reaction, is as low as 30–40%[8]. To promote the selective production of CO, specific catalysts and reaction conditions need to be chosen in order to stabilize the formation of the $CO_2^{\bullet-}$ radical and simultaneously increase the energy barrier of the HER[8,16,17].

Consequently, explicit knowledge of the chemical reactions and equilibria of $CO_2$ in aqueous solutions is essential for advancing carbon dioxide chemistry[8,18–24]. First, gaseous $CO_{2(g)}$ forms an equilibrium with its solvated species $CO_{2(aq)}$. The carbon dioxide concentration $c(CO_{2(aq)})$ in aqueous media at a partial pressure $p_{CO_2}$ is determined by Henry's law,

$$c(CO_{2(aq)}) = H^{cp}(CO_2) \cdot p_{CO_2} \quad (4)$$

$H^{cp}$ denotes the Henry solubility constant of a species for a specific solvent at a given temperature. At 10 °C, the Henry solubility is $H^{cp}(CO_2) = 5.2 \times 10^{-4}$ mol $m^{-3}Pa^{-1}$, resulting in $c(CO_{2(aq)}) = 52.7$ mmol at 1013 hPa $CO_2$ partial pressure[8]. As $CO_{2(aq)}$ is in dynamic equilibrium with bicarbonate ($HCO_3^-$), two possible reaction pathways exist,

$$CO_{2(aq)} + H_2O \underset{k_{-1a}}{\overset{k_{1a}}{\rightleftharpoons}} H_2CO_3 \quad (5)$$

$$H_2CO_3 + H_2O \overset{inst.}{\rightleftharpoons} HCO_3^- + H_3O^+ \quad (6)$$

$$CO_{2(aq)} + OH^- \underset{k_{-1b}}{\overset{k_{1b}}{\rightleftharpoons}} HCO_3^- \quad (7)$$

where $k_i/k_{-i}$ are the respective forward/backward reaction rates. The first pathway comprises a two step reaction in which carbon dioxide first reacts with water to form carbonic acid ($H_2CO_3$) (Eq. (5)). In the second step carbonic acid deprotonates to form bicarbonate (Eq. (6)). This step is several orders of magnitude faster compared to the formation of carbonic acid. The second, direct pathway is a slow one-step reaction of carbon dioxide and hydroxide yielding bicarbonate (Eq. (7)). In calculations the direct path is usually neglected because it becomes relevant only at higher pH values. Finally, $HCO_3^-$ can deprotonate further to carbonate ($CO_3^{2-}$):

$$HCO_3^- + H_2O \underset{k_{-2}}{\overset{k_2}{\rightleftharpoons}} CO_3^{2-} + H_3O^+ \quad (8)$$

The deprotonation of $HCO_3^-$ to $CO_3^{2-}$ is several orders of magnitude faster than the formation of bicarbonate[18,20]. Since acidic protons and alkaline hydroxide ions are involved for all reaction steps, the equilibrium of $CO_2/HCO_3^-/CO_3^{2-}$ strongly depends on the pH value. Combining the equilibrium constants $K_1 = k_{1a}/k_{-1a}$ and $K_2 = k_2/k_{-2}$ with $c_{total} = c(CO_2) + c(HCO_3^-) + c(CO_3^{2-})$ for the sum of all involved carbon species yields the following equilibrium concentrations[22–25]:

$$c(CO_2) = c_{total} \left[ 1 + \frac{K_1}{c(H_3O^+)} + \frac{K_1 K_2}{c^2(H_3O^+)} \right]^{-1} \quad (9)$$

$$c(HCO_3^-) = c_{total} \left[ 1 + \frac{c(H_3O^+)}{K_1} + \frac{K_2}{c(H_3O^+)} \right]^{-1} \quad (10)$$

$$c(CO_3^{2-}) = c_{total} \left[ 1 + \frac{c(H_3O^+)}{K_2} + \frac{c^2(H_3O^+)}{K_1 K_2} \right]^{-1} \quad (11)$$

In the electrolysis of aqueous $CO_2$ solutions it was shown that only solvated $CO_2$ is directly involved in the reduction reaction. Even though $HCO_3^-$ cannot be electrochemically reduced, it can supply $CO_2$ to the electrolysis via the equilibrium reaction. Thus, the pH value of the electrolyte is an essential parameter for optimizing of the electrolytic $CO_2$ reduction. At high pH values, $CO_2$ is only present in low concentrations. However, at low pH values, where the $CO_2$ concentration is maximum, the HER predominates because of the high amounts of $H^+$. Therefore, $CO_2$ electrolysis is usually performed at an intermediate pH range of 7–9, where $HCO_3^-$ is the prevalent species in solution and the concentration of solvated $CO_2$ approaches its solubility limit in water. Then, $CO_3^{2-}$ contributes less than 1% of the total carbon concentration. Noteworthy here is that the electrolysis reaction creates an alkaline environment near the working electrode, whereby the pH and the concentration of carbon species near the electrocatalyst will differ from values of the bulk solution[26].

While the medium pH region is optimal for the electrolytic reduction of $CO_2$, the electrolysis reaction in aqueous media is still limited by its low solubility in water. The reaction rate of the $CO_2/HCO_3^-$ equilibrium and the movement of bulk $CO_2$ towards the electrode are insufficient to replenish $CO_2$ at high current densities. To overcome this challenge, gas diffusion electrodes (GDEs) have been employed. In GDE electrolysis set-ups, $CO_2$ is supplied via the gas phase into the catalytically active pores of the electrode[2,6,8,27]. Maximum current densities of up to 300 mA $cm^{-2}$ have been reported by employing this method[7].

For large scale applications of electrolytic $CO_2$ utilization, currents starting at 400 mA $cm^{-2}$ are required. This necessitates

the optimization of reaction conditions particularly in electrode proximity. However, studies of reaction parameters, such as molecular mobility and chemical exchange rates, are currently limited by the lack of *in operando* measurement methods, i.e., methods applied under catalytic conditions of the electrolysis[28–30]. Only a limited number of *in operando* studies have been performed on electrolytic $CO_2$ reduction in aqueous media, mainly using infrared and Raman spectroscopy[31–34]. NMR techniques and setups have only rarely been employed in this field, although the first *in operando* NMR study was already performed in 1975 by Richards et al.[35–37]. Most *in operando* NMR setups employ special cells and probes, which hinders their applicability and adaptability. Nevertheless, NMR offers an extensive tool set for investigating chemical reactions, as was demonstrated for a wide variety of catalytic, mechanistic, and kinetic studies[38–44]. The method is sensitive to the chemical environment of reaction species and can monitor changes in their molecular mobility as well as chemical exchange phenomena[45,46].

As evident from the dynamic equilibrium reactions in eqs. (5) to (7), the study of chemical exchange via NMR is of particular interest for the $CO_2$ electrolysis. For a system of two exchanging species, a NMR experiment depends on the exchange rate, i.e., the inverse of the exchange time constant $T_{exc}$, and the difference in the respective resonance frequencies $|\Delta\nu|$. If the exchange rate is significantly lower than this difference, two distinct signals are present in the NMR spectrum, and the dynamic chemical equilibrium can be qualitatively and quantitatively assessed by 1D or 2D Exchange Spectroscopy (EXSY)[47,48]. If the exchange rate is significantly higher than the difference in resonance frequencies, one averaged, sharp signal is observed, and exchange may be studied by relaxation experiments. In between these two boundary cases, the individual signals of the exchanging species start to broaden and merge, until eventually coalescing into one broad peak that narrows as the exchange rate increases further. For the case of equimolar concentration, coalescence takes place at an exchange rate $r_{exc.}^{coalesc}$ given by[48]

$$k_{exc}^{coalesc} = \frac{|\pi\Delta\nu|}{\sqrt{2}}. \tag{12}$$

As evident, the coalescence point depends on $|\Delta\nu|$ and thus the the main magnetic field strength $B_0$, which can be exploited to identify exchange processes. Furthermore, using the modified Bloch equations developed by Gutowsky and Holm, exchange spectra can be calculated at various $B_0$ field strengths[49]. By comparing these calculated spectra with observations, the experimental exchange rate can be estimated.

For the $CO_2$ reduction reaction, two types of exchange must be considered. The first one involves the equilibrium of $CO_2$, $HCO_3^-$ and $CO_3^{2-}$ in aqueous media. At typical electrolysis conditions, the exchange of $CO_2$ and $HCO_3^-$ is slow enough to be studied via EXSY. The equilibrium between $HCO_3^-$ and $CO_3^{2-}$ is fast even at low temperatures, therefore an averaged carbonate peak is expected in most studies. The position of this peak may be used for *in operando* pH determination[50,51]. The second equilibrium of interest appears at high ion concentrations found in most electrolytes. Then, a solution with exclusively free, non-interacting ions can no longer be assumed, and ion pairing may take place. Several kinds of ion pairs are described in literature, which are part of a dynamic equilibrium with free ions[52]:

$$FI \rightleftharpoons 2SIP \rightleftharpoons SIP \rightleftharpoons CIP \tag{13}$$

Going from left to right, the distance between the cation and anion decreases. Free ions (FI), possessing individual solvation shells, are spatially separated from their respective counter ions. In the case of solvent separated ion pairs (2SIP), the solvation shell of each distinct ion is still intact, but their respective solvent

molecules directly contact one another. For solvent shared ion pairs (SIP), a single solvation layer is shared between anion and cation. Finally, there is no solvent layer between the ions of contact ion pairs (CIP), where anion and cation are in direct contact with one another. For bicarbonate electrolytes, which are frequently used in the $CO_2$ reduction reaction, the effects of ion pairing have not been fully understood.

This work aims to elucidate how a potential dependent change in electrolyte chemistry affects the aqueous $CO_2$ equilibria by means of *in operando* NMR spectroscopy. A previously published setup is used that employs a commercial NMR probe and allows for $^{13}C$ NMR experiments with good sensitivity and high resolution[53]. The electrolytic $CO_2$ reduction is studied at several potentials using a silver working electrode. The active species are tracked during electrolysis at two different $B_0$ fields using $^{13}C$ and $^{23}Na$ NMR. Changes in molecular dynamics are assessed using longitudinal and transverse relaxation time constants. Exchange experiments are performed to determine rate constants of the dynamic equilibrium between slowly exchanging species. Eventually, these results are combined in a model that describes the potential dependence of the steady-state solvation dynamics in the electrolyte.

## Results and discussion

**Electrochemistry and general evolution of $^{13}C$ spectra**. The $^{13}C$ NMR spectrum of the $CO_2$ saturated electrolyte solution is shown in Fig. 1. It consists of a $CO_2$ signal at 125.9 ppm and a coalesced $HCO_3^-/CO_3^{2-}$ signal at 161.8 ppm. At the present pH of 8.2 the $HCO_3^-/CO_3^{2-}$ equilibrium is expected to consist 99% of bicarbonate. At the start of the *in operando* experiment, the solvated $CO_2$ concentration has been determined as 55 mmol $L^{-1}$, and total carbon ($CO_2 + HCO_3^- + CO_3^{2-}$) as 1.87 mol $L^{-1}$, using an external reference experiment. The shape of all $^{13}C$ signals consist of a sharp peak and a downfield shoulder, which is due to $B_0$ distortions in proximity to the metal electrodes. The sharp peak component of $HCO_3^-$ is assumed to be a coalesced signal of $HCO_3^-$ present as free ions and ion pairs, as shown later.

During the Open Circuit Voltage (OCV) stage, i.e., when no current is applied to the cell and the system is allowed to relax, the equilibrium potential drops from $-31$ mV to $-42$ mV vs. Ag/AgCl. The peak shapes of all $^{13}C$ signals remain constant, and only small downfield frequency shifts in the single digit ppb range are observed during OCV. The integral of the $CO_2$ signal,

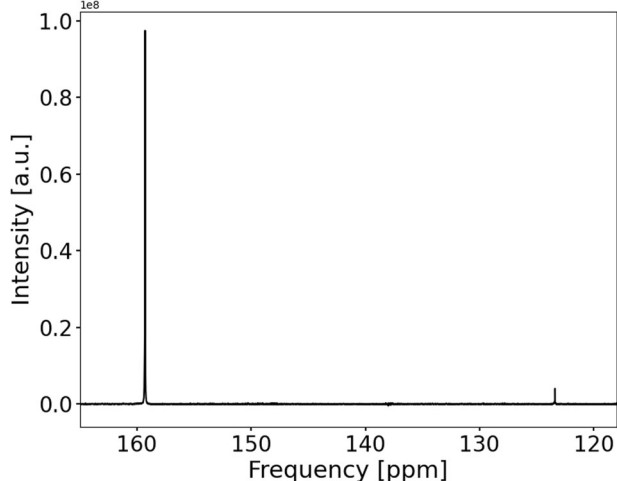

**Fig. 1 $^{13}C$ spectrum of a $CO_2$ saturated 1 mol $L^{-1}$ KHCO₃ electrolyte.** The spectrum consists of the solvated $CO_2$ signal at 125.9 ppm and the coalesced $HCO_3^-/CO_3^{2-}$ signal at 161.8 ppm.

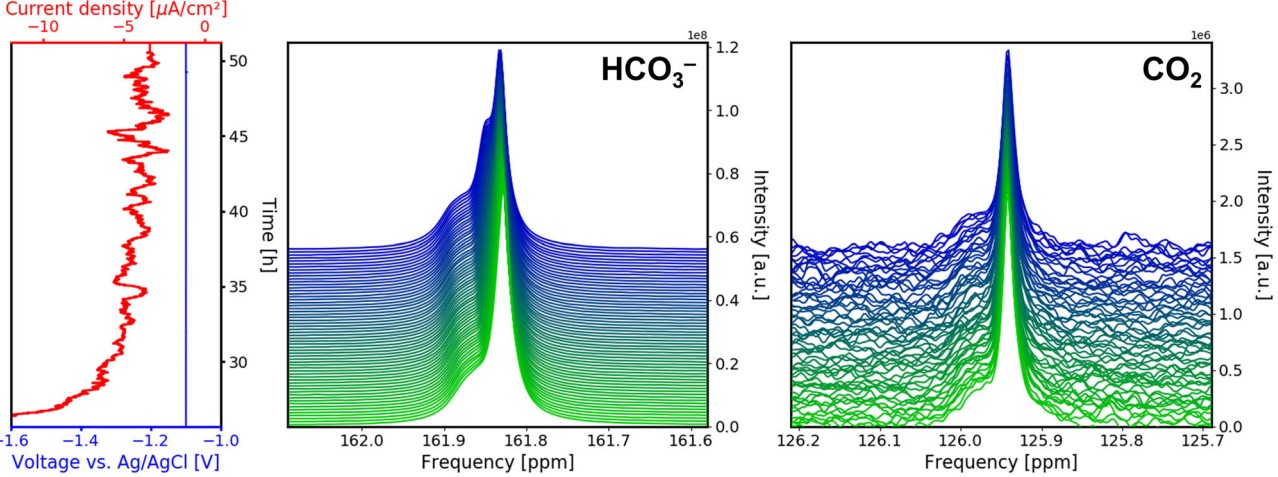

**Fig. 2 Overview of $^{13}$C signal evolution during the chronoamperometry (CA) stage.** Time evolution of voltage and current density as well as the corresponding $^{13}$C NMR signals at $B_0 = 14.1$ T of $HCO_3^-$ and $CO_2$ during CA with a constant voltage of -1.1 V. A new peak emerges for the $HCO_3^-$ signal.

however, decreases significantly to 62% (34.1 mmol L$^{-1}$) over the course of 25.6 h of OCV, which has been explained as an effect of a shifting carbonate equilibrium, and is supported by the change in equilibrium potential[53].

For the Chronoamperometry (CA) stage, a constant potential of $-1.1$ V vs. Ag/AgCl was applied, and the evolution of current as a function of time was observed. The current density at the beginning of this stage was negative at $-12\,\mu\text{A cm}^{-2}$, but increases quickly to values around $-4\,\mu\text{A cm}^{-2}$ (Fig. 2 left) at *ca.* 35 h. This change in current density is caused by an interplay of multiple contributing factors. These factors include the depletion of the reaction educt $CO_2$ in electrode vicinity as well as the formation of gas bubbles, which can reduce the effective electrode area upon adhesion. Another significant contribution to the evolution of the current density is the vertical orientation of the electrodes. Regions of the WE and CE closest to each other require slightly less voltage for electrolysis to take place due to internal resistance of the electrolyte solution. i.e. these regions experience a smaller iR drop. At the start of electrolysis, educt is sufficiently supplied in proximity to these low iR drop regions, and current density is considerably negative. However, after $CO_2$ has been depleted, the electrolysis reaction shifts place to regions further apart with a higher iR drop. This causes the current density to increase to less negative values, until an equilibrium state has been reached. In addition, the current density strongly fluctuates during the whole CA experiment. These fluctuations are assumed to arise from a combination of measurement uncertainties due to the low current in the nanoampere range and increased noise levels due to the interaction of the electric leads with the RF pulses of the NMR experiment as well as bubble formation and release.

During the CA stage, the $HCO_3^-$ signal undergoes a transformation as the sharp peak component at 161.8 ppm splits into two components with a separation of 2.9 Hz at a magnetic field of 14.1 T (Fig. 2 middle). The peak separation is assumed to be linked to a decreasing exchange rate between free $HCO_3^-$ and the anions forming an ion pair, which is discussed in detail further below. By contrast, the $CO_2$ signal does not change in signal shape or position, but decreases in intensity from 62 % of its original value to 37% at the end of the CA stage, i.e. the $CO_2$ concentration decreases from 34.1 mmol L$^{-1}$ to 20.4 mmol L$^{-1}$ (Fig. 2 right). Assuming a faradaic efficiency of 100%, the $CO_2$ reduction reaction consumes less than 0.28 mmol L$^{-1}$ of educt. Thus the additional decrease in $CO_2$ concentration of 13.4 mmol has to be caused by a shift of the equilibrium of $CO_2$ in aqueous

media and further a shift of local pH in the sensitive NMR volume due to the electrolysis reaction. Using the given concentrations of carbon species, a local pH of 8.4 was calculated via Eqs. (9), (11), which is in line with the amount of consumed protons or accordingly formed hydroxid given the flown charge eq. (3). This, however, implies that exchange between electrode compartments is slow at least for charged species such as $OH^-$ and $H^+$.

For the Chronopotentiometry (CP) stage at the end, a constant current density of $-10\,\mu\text{A cm}^{-2}$ was applied to the cell (Fig. 3 left), and the evolution of the potential versus time was recorded. At the beginning of this stage, the potential rapidly decreases to $-1.44$ V vs. Ag/AgCl, and increases afterwards until it reaches an equilibrium value of $-1.41$ V vs. Ag/AgCl at *ca.* 65 h. This overshoot of the overpotential could to be caused by turbulent micro flow currents[54]. Microflow currents are formed to compensate the sudden formation of density gradients in the electrolyte during the onset of the electrolysis reaction and double layer formation, and decrease in velocity after equilibration.

The evolution of the $^{13}$C signal during the CP stage continues the trends observed during CA. Separation between the split $HCO_3^-$ peak components increases to 4.7 Hz at 14.1 T (Fig. 3 middle). This is due to a further decrease in exchange rate between $HCO_3^-$ existing as free ion and in an ion pair. The peak shape and position of $CO_2$ continue to remain unchanged, and the intensity further decreases to 22% of its original value, i.e. a concentration of 12.1 mmol L$^{-1}$, at the end of the CP stage (Fig. 3 right). For the electrolysis reaction, a maximum of 0.50 mmol L$^{-1}$ of the initial $CO_2$ concentration is consumed, thus a further decrease of the $CO_2$ concentration is likely from an increase in pH value in electrode proximity. Using the $CO_2$ in water equilibrium reactions (Eq. (9)–Eq. (11)), the pH value at the end of CP is calculated to be 8.5. It is important to note that the pH value calculated from the $CO_2$ concentration is an averaged value of the electrolyte inside the sensitive volume. It is a past and current point of research that a pH gradient is formed during $CO_2$ electrolysis, which plays a significant role in catalyst activity and selectivity[50,51].

**Evolution of the electrolyte ion signals**. As stated above, the $^{13}$C $HCO_3^-$ signal splits into two peaks after OCV, which increasingly separate when an external potential is applied. The separation can be illustrated using peak deconvolution by peak fitting as shown in Fig. 4 for the 14.1 T magnetic field. This behavior is reminiscent of a system exchanging between two

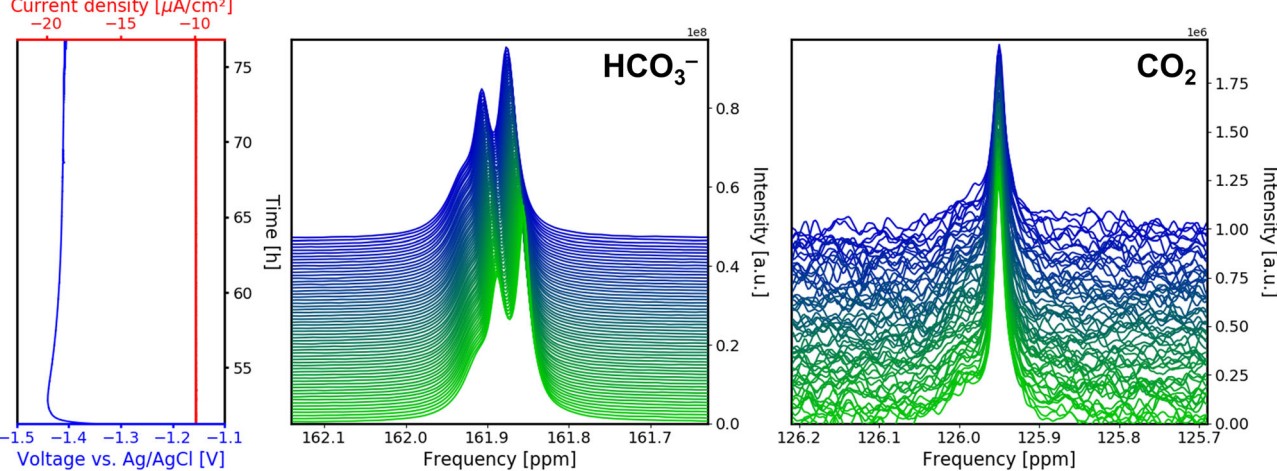

**Fig. 3 Overview of $^{13}$C signal evolution during the chronopotentiometry (CP) stage.** Time evolution of voltage and current density as well as the corresponding $^{13}$C NMR signals at $B_0 = 14.1$ T of $HCO_3^-$ and $CO_2$ during CP with a constant current density of -10 $\mu$A cm$^{-2}$. The separation between the $HCO_3^-$ signal components increases during this stage.

environments, where exchange occurs at a rate on the order of the NMR frequency differences. It is important to note that in the case of $HCO_3^-$, the two exchanging environment must be magnetically similar as the difference in chemical shift of bicarbonate environments is small. It can also be noted that the splitting cannot be caused by a decelerated $HCO_3^-/CO_3^{2-}$ exchange, as the chemical shift difference between both species is significantly larger. Thus, it is proposed that the two environments are different forms of $HCO_3^-$ in solution. In diluted solutions, cations and anions exist usually as free ions, i.e. an ion which is fully solvated by solvent molecules and mostly unaffected by other ions. For solutions of higher concentration, ion pairs exist in addition to free ions. Between the three types of ion pairs discussed in literature, the contact ion pair (CIP) is the most unlikely one observed in the presented $^{13}$C spectra, as the direct contact of $HCO_3^-$ with its cation is expected to have a larger effect on its electron sphere and thus the chemical shift. However, the experimental NMR data is not sufficient to distinguish between solvent shared ion pairs (SIP) and solvent separated ion pairs (2SIP); it may even be the case that both forms co-exist, with rapid exchange between them. Thus, the term solvent associated ion pairs (xSIPs) is proposed in this manuscript, where x implies one or two solvent layers, as both SIPs and 2SIPs share the feature that anion and cation are associated by solvent molecules.

The upfield component in Fig. 4b, c is assigned to the free $HCO_3^-$ anions. The downfield component is assigned to $HCO_3^-$ xSIPs, as the close proximity of bicarbonate to its counter cation is expected to decrease the electron density of the anion. These assignments were validated using $^{13}$C and $^{23}$Na DFT chemical shift calculations. It was found that the exact geometry of the hydration sphere of each ion in the simulation has generally a larger impact on the chemical shift than the mutual influence of anion and cation at distances larger than a contact ion pair. Thus, two hydration sphere models of varying complexity were applied. In the first model, the hydration sphere of a contact ion pair was geometry optimized. Then, the anion and cation were pulled apart with their respective hydration spheres intact, i.e. the individual ion hydration spheres are static throughout the simulations at various ion distances (Fig. S19 in the SI). Using this static hydration sphere geometry, the $^{13}$C and $^{23}$Na chemical shifts were calculated at various sodium–bicarbonate distances (Fig. S20 in the SI).

The second model included a molecular dynamics (MD) model of the hydration sphere. Here, conformational dynamics of the

hydration spheres were sampled by MD simulations of the ion pairs at multiple constrained distances. For each of the samples, the $^{13}$C and $^{23}$Na chemical shifts were calculated, and their average values were computed afterwards (Fig. S21 and Table S1 in the SI).

Both models are in agreement with the peak assignment both of the $^{13}$C spectra and the $^{23}$Na spectra, which are presented later in the text. It can be highlighted that the simulations not only predict the direction of the shift in chemical shift correctly, but also the order of magnitude, although no further statement on the presence of 2SIPs versus SIPs can be given. Further discussion on the chemical shift calculations are given in the supplementary methods section in the SI.

In the experimental spectra, free ion and xSIPs signals are separated when the exchange rate between these ion forms is slower than the coalescence point as described in Eq. (12), and they are coalesced into a single peak (Fig. 4a) if it is faster. As evident from the evolution of the $HCO_3^-$ signal, the exchange rate is fast under OCV conditions, and continuously decreases when an increasingly negative potential is applied. Thus, it can be stated that stable electrolyte ions and ion pairs, i.e. species with a long life time, are formed when a potential is applied to the system.

The presence of two exchanging environments for the electrolyte ions can be further verified by two experiments. First, the $^{13}$C in operando spectra can be recorded at a lower magnetic field strength. The coalescence point is dependent on the $B_0$ field, as according to Eq. (12) it is a function of the absolute peak separation in Hz, and not the relative one in ppm. Thus, under otherwise identical conditions, a lower peak separation of the $HCO_3^-$ signal is expected at lower field strength. This is proven true as evident in Fig. 5. Analogous to the in operando experiments at 14.1 T, one single, coalesced $HCO_3^-$ is observed at OCV (Fig. 5a), which splits up into free ion and xSIP signal for the CA and CP stage. In accordance with the predictions, the peak separation is only 1.0 Hz (Fig. 5b) and 1.4 Hz (Fig. 5c) at the end of the CA and CP stage, respectively, and thus significantly lower compared to the experiment at 14.1 T.

Furthermore, based on the experimental data at 14.1 T, exchange spectra at 9.4 T can also be calculated using the modified Bloch equations given in the work of Williams et al.[49]. By comparing these calculations with the observed spectra at 9.4 T, the global exchange rate of the system can be estimated to be 15 s$^{-1}$. This is the minimum exchange rate at CP, and increases

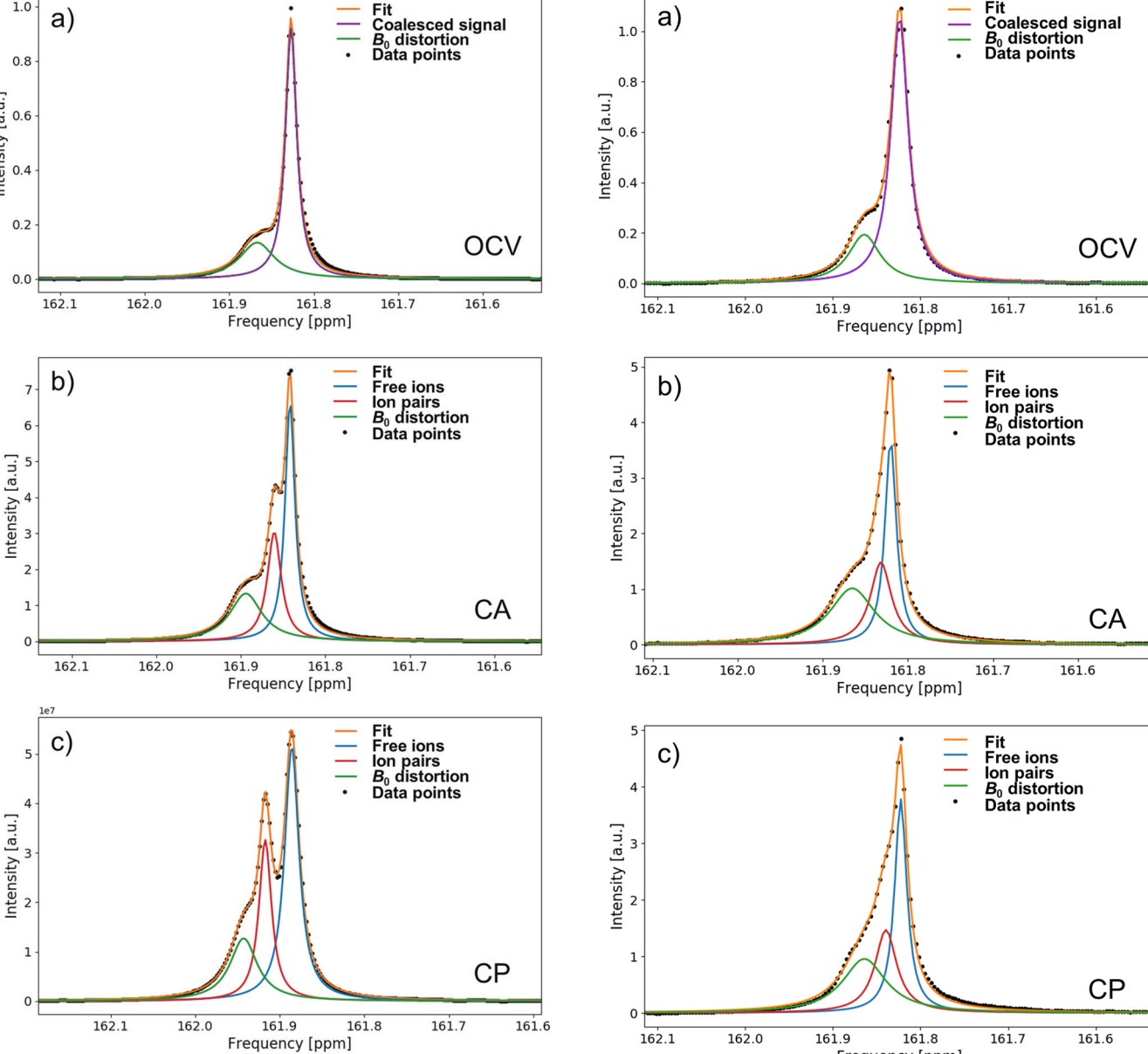

**Fig. 4 Signal deconvolution of the $^{13}C$ $HCO_3^-$ NMR signal at 14.1 T (150.9 MHz resonance frequency).** The $HCO_3^-$ evolves from the open circuit voltage (OCV) stage **a** over the chronoamperometry (CA) stage **b** to the chronopotentiometry (CP) stage **c**. Two signal components are assigned to $HCO_3^-$ existing as free ions (blue) and in an ion pair (red). For high exchange rate between these two states, i.e., during OCV, the two signal components coalesce into a single peak (violet). An additional signal component (green) represents the $B_0$ distortion in proximity to the working electrode.

**Fig. 5 Signal deconvolution of the $^{13}C$ $HCO_3^-$ NMR signal at 9.4 T (100.6 MHz resonance frequency).** The $HCO_3^-$ evolves from the open circuit voltage (OCV) stage **a** over the chronoamperometry (CA) stage **b** to the chronopotentiometry (CP) stage **c** in a similar fashion to Fig. 4, where the free ion are depicted in blue, ion pairs in red, and coalesced signal in purple. An additional signal component (green) represents the $B_0$ distortion in proximity to the working electrode. Due to the weaker $B_0$ field strength, the separation of the free ion and ion pair signal is not as pronounced.

during the CA and OCV stage. The global exchange rate is the inverse of the inverse exchange time, a combined value of forward and backward exchange times $\tau_{global}$, which is defined as $\tau_{global} = \tau_{forward} \cdot \tau_{backward}/(\tau_{forward} + \tau_{backward})$.

For the second verification experiment, *in operando* spectra were recorded for the counter ion, which is potassium or sodium, depending on the bicarbonate salt used. As $^{23}Na$ has a significantly higher NMR sensitivity compared to $^{39}K$, $^{23}Na$ *in operando* experiments were conducted. Due to the inherently broader lines of $^{23}Na$ caused by its short $T_2$ time constants, the evolution of the sodium cation signal is not as obvious, but observable nonetheless (Fig. 6). The component originating from

the $B_0$ distortion cannot be distinguished, thus the $^{23}Na$ signal consists of a signal peak at OCV. At the CA stage, the formation of a shoulder can be clearly observed, which becomes more pronounced during the CP stage. A precise value for the peak separation cannot be reliably determined for $^{23}Na$, as the peak deconvolution is challenging given the high line widths. However, the peak separation can be estimated to be higher than 6 Hz, which differs significantly from the peak separation for the $^{13}C$ $HCO_3^-$ signal. Thus, it can be safely established that the evolution of the electrolyte signals is not due to distortion of the $B_0$ field, but must be induced by physico-chemical processes taking place in the electrolyte solution.

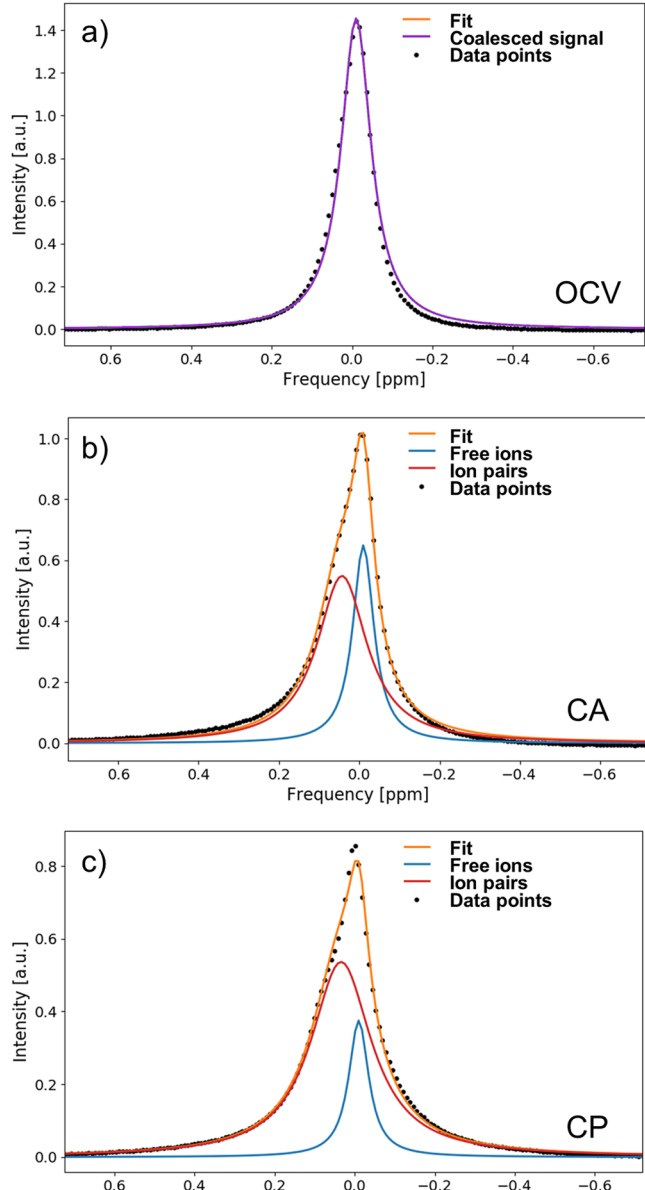

**Fig. 6 Deconvolution of the $^{23}$Na NMR resonance caused by Na$^+$ at 14.1 T (158.7 MHz resonance frequency).** The Na$^+$ cation signal evolves from a coalesced state (purple) in the open circuit voltage (OCV) stage **a** over the chronoampoerometry (CA) stage **b** to the chronopotentiometry (CP) stage **c** by splitting into the two components of free ions (blue) and ion pairs (red), similar to the HCO$_3^-$ signal in Fig. 4. However, due to the inherently broader line widths of $^{23}$Na, the peak splitting is not as pronounced and the $B_0$ distortion artifact cannot be observed separately.

**Effects on electrolyte chemistry.** The effect of the formation of stable free ions and ion pairs can be studied by investigation of NMR relaxation times as well as the exchange rate of the dynamic equilibrium between CO$_2$ and HCO$_3^-$ in aqueous solution. As presented in Table 1, it is obvious that the CO$_2$/HCO$_3^-$ exchange rate increases during the CA and CP stage. Therefore, the CO$_2$/HCO$_3^-$ equilibrium reaction accelerates when the exchange between free ions and xSIPs slows down, and more stable solvated ions and ion pairs are formed.

The origin of this effect can be explained based on the longitudinal ($T_1$) NMR relaxation time constants presented in Fig. 7 and Table 2. Here, after the coalesced HCO$_3^-$ peak splits

**Table 1 Exchange time constants $T_{exc}^{CO_2/HCO_3^-}$ for the equilibrium reaction between bicarbonate and carbon dioxide for each experimental stage at 10 °C (14.1 T), obtained from eq. (14).**

| Stage | OCV | CA | CP |
|---|---|---|---|
| $T_{exc}^{CO_2/HCO_3^-}$ [s] | 2.86 ± 0.26 | 2.19 ± 0.33 | 1.56 ± 0.20 |

The stages are open ricuit voltage (OCV), chronoamperometry (CA), and chronopotentiometry (CP).

**Fig. 7 Evolution of the $^{13}$C $T_1$ relaxation time constants for CO$_2$ (green circles) and HCO$_3^-$ (purple, red, and blue circles), starting at the open circuit voltage (OCV) stage.** At the chronoamperometry (CA) stage, the HCO$_3^-$ signal splits into two components for which different relaxation time constants for free ions and xSIPs can be distinguished. With increasingly negative potential at the CA and chronopotentiometry (CP) stage, CO$_2$ approaches the HCO$_3^-$ xSIPs in terms of $T_1$. The error bars represent the fitting error received for $T_1$ determination. The term xSIP represents ion pairs with one ($x = 1$) or two ($x = 2$) solvent layers between anion and cation.

into two components during the CA stage, $T_1$ of free ions and xSIPs differ significantly, and the gap increases during the CP stage. $T_1$ of the xSIPs is lower than of the free ions, which hints towards a longer correlation time and thus lower mobility of the xSIPs according to Bloembergen-Purcell-Pound (BPP) relaxation theory[55]. This is apparent, as an ion pair of HCO$_3^-$ and Na$^+$ has a larger inert mass and hydrodynamic radius compared to a free anion or cation. In addition, $T_1$ of HCO$_3^-$ in xSIPs may be affected by the quadrupolar moment of sodium or potassium due to their close proximity. An analogous effect can be observed for the transverse ($T_2$) NMR relaxation time constants of the HCO$_3^-$ components.

The relaxation time constants $T_1$ and $T_2$ of CO$_2$ also surprisingly decrease as a function of the applied potential. This cannot be an effect of the CO$_2$ reduction reaction, as only a small percentage of CO$_2$ reacts during each stage. Furthermore, only CO$_2$ in electrode proximity should be affected; NMR, however, measures the bulk of the aqueous electrolyte solution. Instead, $T_1$ of CO$_2$ seems to approach that of the xSIP HCO$_3^-$ species due to the increasing exchange rate between CO$_2$ and HCO$_3^-$. This is especially apparent for the CP stage, where the relaxation time constants of CO$_2$ fall below that of free HCO$_3^-$. The observation implies that the dynamic equilibrium of CO$_2$ and HCO$_3^-$ in aqueous solution preferentially takes place with HCO$_3^-$ in ion pairs, and to a lesser extent with free bicarbonate. In combination

with the observation of increased $CO_2/HCO_3^-$ exchange rates, this suggests a model where the xSIP of $HCO_3^-$ and its cation catalyzes the (de-)hydration of $CO_2$ in aqueous solution. A description of the catalyzed equilibrium reaction in case of a sodium cation is depicted in Fig. 8. Here, the cation stabilizes the negative charge distribution at two oxygen atoms of $CO_2$ or $HCO_3^-$, and thus aids the addition or elimination of a hydroxide group. As a sufficient life time of the catalyst is critical for the completion of the catalytic cycle, its efficiency increases as the exchange rate between free ions and xSIPs decreases, i.e. the xSIPs become more stable. A similar mechanism has been found for the carbonic anhydrase enzyme, which regulates the pH value in human bodies. There, the catalytic center is a zinc cation that binds to the oxygen atoms of $CO_2$ or $HCO_3^-$. The complex and specialized nature of an enzyme enables significantly higher efficiency and reaction rates. However, the catalytic reactions remains analogous.

To conclude, a catalytic activity of the electrolyte cation for the $CO_2/HCO_3^-$ equilibrium reaction is suggested based on the evolution of relaxation and exchange times akin to enzymatic systems found in nature. The catalytic activity increases with the life time of the cation in a xSIP, which is inversely proportional to the exchange rate between xSIPs and free ions (Fig. 9).

As a last note, the relaxation time constants of $HCO_3^-$ for the component which experiences $B_0$ distortion reveal that the dynamics of the electrolyte volume in electrode proximity differ significantly from the bulk solution. This is especially noticeable for $T_1$, which is identical at OCV for the bulk and electrode proximate signal components. Only when a potential is applied at the CA and CP stages, the relaxation times change from the

values found in the bulk electrolyte. These changes are unlikely caused by $B_0$ and $B_1$ field distortions, as the relaxation times are not sensitive to ppm changes in the main magnetic field, and the saturation recovery and CPMG NMR pulse sequences used here are assumed to be robust in terms of $B_1$ homogeneity. Further investigations of the electrolyte regions close to the electrocatalyst are currently ongoing.

**Origin of the electrolyte ion separation**. Upon application of an external potential, the changes to the electrolyte chemistry described here represent a separation of electrolyte ions into ion pairs and free ions. This effect can be interpreted as a consequence of Le Chatelier's law. Upon application of the potential, an electric field is generated between the electrodes. The charged electrodes attract cations or anions from the solution in order to form a double layer. Thus, the local charge density of the electrolyte increases in proximity to the electrodes, which is counteracted by two mechanisms. First, the shielding of the ions by the formation of isolating solvent shells is more pronounced, which leads to the formation of stable free ions. Secondly, stable ion pairs are formed to combat the concentration gradient and charge separation taking place in the solution. As a result of both mechanisms, the exchange rate between the stabilized free ions and ion pairs is decreased.

A significant consequence of the observations is that the electric field applied during electrolysis affects the bulk of the electrolyte. Otherwise, only minuscule observations would be made, as NMR is an analytical method with bulk sensitivity. This notion is opposed to the classical models in electrochemistry, e.g. the Stern model of the electrical double layer, where the effect of the applied potential only extends up to a few nanometers from the electrodes into the electrolyte, and can be approximated by the Debye length. This observation is consistent with a recent publication of Zhang et al., which also suggests that the chemical impact of the electric field in electrolysis may extend significantly into the bulk of the electrolyte[56].

**Table 2 Longitudinal and transverse $^{13}$C relaxation time constants $T_1$ and $T_2$ of $HCO_3^-$ signal components during each experimental stage at 10 °C (14.1 T).**

| Stage | Species | Component | $T_1$ [s] | $T_2$ [s] |
|-------|---------|-----------|-----------|-----------|
| OCV | $CO_2$ | Full signal | 13.18 ± 0.71 | 2.15 ± 0.25 |
| | $HCO_3^-$ | Coalesced signal | 11.73 ± 0.05 | 0.81 ± 0.01 |
| | | $B_0$ distortion | 11.75 ± 0.40 | 0.69 ± 0.03 |
| CA | $CO_2$ | Full signal | 12.39 ± 0.76 | 1.08 ± 0.13 |
| | $HCO_3^-$ | Free ions | 11.52 ± 0.09 | 1.00 ± 0.01 |
| | | xSIPs | 10.69 ± 0.10 | 0.79 ± 0.02 |
| | | $B_0$ distortion | 12.96 ± 0.15 | 1.05 ± 0.02 |
| CP | $CO_2$ | Full signal | 11.43 ± 1.41 | 0.78 ± 0.17 |
| | $HCO_3^-$ | Free ions | 13.91 ± 0.13 | 1.34 ± 0.01 |
| | | xSIPs | 10.95 ± 0.12 | 0.93 ± 0.02 |
| | | $B_0$ distortion | 13.06 ± 0.35 | 1.19 ± 0.03 |

The error represents the fitting error received for $T_1$ and $T_2$ determination. The term xSIP represents ion pairs with one ($x = 1$) or two ($x = 2$) solvent layers between anion and cation.

## Conclusion

*In operando* $^{13}$C and $^{23}$Na NMR was applied to follow the major evolution taking place for the electrolyte anion bicarbonate and the respective cation during electrolytic $CO_2$ reduction. As a function of increasingly negative potential, the exchange rate between free electrolyte ions and electrolyte solvent associated ion pairs (xSIPs) decreased. As a result, distinct chemical environments with longer lifetime were formed for both $HCO_3^-$ and its counter ion. The formation of xSIPs with a long lifetime may accelerate the equilibration reaction between $HCO_3^-$ and $CO_2$, which promotes the resupply of $CO_2$ during the electrolytic reaction based on a mechanism were the cation stabilizes the the angled structure and negative charge of $HCO_3^-$ during deprotonation.

**Fig. 8 Proposed reaction scheme of the bicarbonate dehydration.** The reaction is catalyzed by the cation of a xSIP, exemplary shown for Na$^+$. A similar catalytic reaction is known from the enzyme carbonic anhydrase. The term xSIP represents ion pairs with one ($x = 1$) or two ($x = 2$) solvent layers between anion and cation.

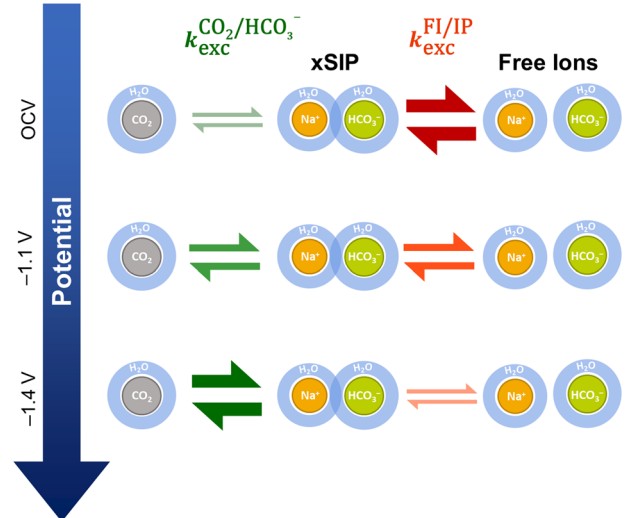

**Fig. 9 Correlation of equilibrium reaction rates during electrolysis.**
Evolution and correlation of the $CO_2/HCO_3^-$ equilibrium reaction rate
($k_{exc}^{CO_2/HCO_3^-}$) and the exchange rate between free electrolyte ions and xSIPs
($k_{exc}^{FI/IP}$) as a function of potential. With increasingly negative potential, it
was found that $k_{exc}^{FI/IP}$ decreases, while $k_{exc}^{CO_2/HCO_3^-}$ increases simultaneously.
Using $T_1$ relaxation time experiments, it was shown that $CO_2$ preferably is
formed from $HCO_3^-$ that is composing a xSIP, which suggests a catalytic
activity of the electrolyte cation, e.g. $Na^+$. The term xSIP represents ion
pairs with one ($x = 1$) or two ($x = 2$) solvent layers between anion and
cation.

Finally, the change in electrolyte chemistry may impact the
$CO_2$ reduction reaction. The electrolytic conversion of $CO_2$ is
limited by the low solubility in water. Multiple approaches have
been studied to overcome this limitation, of which the most
successful one has been gas diffusion electrodes. But at current
densities >300 mA cm$^{-2}$, as required for an industrial application
of electrolytic $CO_2$ reduction, the solubility of $CO_2$ becomes again
a limiting factor. In theory, most electrolytes have stored a surplus
of $CO_2$ as $HCO_3^-$. However the exchange reaction of the
dynamic equilibrium is too low to be feasible for $CO_2$ resupply
even at medium current densities. This study, however, suggests
that the exchange rate can be manipulated for electrolysis
application, e.g. by tuning the catalytic properties of the electro-
lyte cations. Further research in this field may lead to the design
of special electrolyte cations or co-catalysts that enable $HCO_3^-$ in
the electrolyte to become a more effective supplier of $CO_2$ and
thereby enable higher current densities for the $CO_2$ reduction
reaction.

## Methods
***In operando* electrolysis setup**. $^{13}C$ and $^{23}Na$ NMR was
employed in order to evaluate all the species present during the
electrolysis experiments. For $^{13}C$ *in operando* NMR measure-
ments, an 1 mol L$^{-1}$ aqueous solution of 98% $^{13}C$ enriched
KHCO$_3$ (Sigma Aldrich, Munich, Germany) was used as elec-
trolyte. Potassium salt electrolytes are beneficial for the electro-
lysis performance due to the size of the cation, but the stable
potassium nuclei possess a low sensitivity in NMR spectroscopy.
Thus $^{23}Na$ experiments were performed for the investigation of
the electrolyte cation using an 1 mol L$^{-1}$ aqueous solution of
99.9% pure NaHCO$_3$ (Sigma Aldrich, Munich, Germany) as
electrolyte. The electrolyte was pre-chilled inside a polyethylene
vial in a 10 °C water bath. *Ca.* 1 mL of chilled electrolyte was filled
into a 5 mm NMR tube and bubbled with 99% $^{13}C$ enriched $CO_2$
(Cambridge Isotope Laboratories, Tewksbury, USA) using a 1/16

inch PEEK tube at a flow rate of *ca.* 0.3 mL s$^{-1}$ and a temperature
of 10 °C. Afterwards, the electrode setup was placed inside the
5 mm tube with $CO_2$ saturated electrolyte. The gas phase was
aerated with $^{13}C$ labeled $CO_2$ gas prior to sealing the cell. All
preparation steps were performed at ambient conditions.

The electrolysis cell consisted of a three-electrode setup with a
silver plate working electrode (GoodFellow, Hamburg, Germany),
an Ag/AgCl micro reference electrode made in-house, and a
carbon counter electrode (GoodFellow, Hamburg, Germany). The
three-electrode setup was inserted into a 5 mm NMR tube. The
cell was placed in a commercial NMR probe and connected to a
BioLogic SP-200 potentiostat (BioLogic Science Instruments,
Seyssinet-Pariset, France). Additionally, shielding and noise
reduction equipment were required to achieve serviceable
signal-to-noise ratios for NMR experiments. The setup employed
in this study is described in more detail in our previously
published work[53].

**General NMR experimental procedure**. $^{13}C$ NMR experiments
were performed using either a Bruker Avance III HD spectro-
meter equipped with a 14.1 T magnet (150.9 MHz $^{13}C$ frequency)
or with a 9.4 T magnet (100.6 MHz $^{13}C$ frequency). Both spec-
trometers were equipped with double resonance broad band
probe heads. $^{23}Na$ studies were conducted on the 14.1 T spec-
trometer (158.7 MHz resonance frequency). All NMR experi-
ments were performed at a sample temperature of 10 °C. All
magnets were shimmed until linewidths of *ca.* 1 Hz were achieved
for the $^{13}C$ measurements or *ca.* 30 Hz for $^{23}Na$. For the $^{13}C$
experiments at 14.1 T, a 90° pulses length of 15.5 μs at a pulse
power of 58.7 W with a relaxation delay of 85 s was set. For the
$^{13}C$ experiments at 9.4 T, a 90° pulses length of 14.0 μs at a pulse
power of 39.0 W with a relaxation delay of 85 s was set. For $^{23}Na$
experiments at 14.1 T, 90° pulses with a length of 16.5 μs and
pulse power of 40.0 W, and a relaxation delay of 1 s were used.

***In operando* experimental procedure**. For both the $^{13}C$ and $^{23}Na$
studies, the *in operando* investigations were divided into three
stages. Due to different experiment duration for each nucleus, the
total time of each stage was 25.6 h for $^{13}C$ and 12.3 h for $^{23}Na$
investigations. During the first stage, the system was allowed to
relax at open circuit voltage (OCV). During the second stage,
chronoamperometry (CA) was performed by applying a potential
of − 1.1 V *vs.* Ag/AgCl to the working electrode. During the third
stage, chronopotentiometry (CP) was conducted, where a current
of − 1 μA was applied for the $^{13}C$ experiments resulting in a
specific current density of − 10 μA cm$^{-2}$. The change to a sodium
electrolyte for the the $^{23}Na$ investigations affects the $CO_2$ elec-
trolysis. Thus, the current density was adjusted in order to obtain
similar potentials compared to the $^{13}C$ experiments with potas-
sium electrolyte. Low current densities were employed during the
electrochemical experiments to reduce the formation rate of CO
and H$_2$ gas, which can disturb the electrochemical and the NMR
measurements.

At the beginning of each stage, the system was allowed to settle
in for 12 h while spectra were recorded continuously using a time
step of 6 min for a total of 120 spectra. Full $^{13}C$ and $^{23}Na$ spectra
during OCV, CA, and CP stage for each set of experiments at a
time resolution of 1 hour are presented in the SI figs. S1–S9. For
the $^{13}C$ experiments, the $CO_2$ saturated electrolyte was monitored
by determining the longitudinal relaxation time constant $T_1$, the
transverse relaxation time constant $T_2$, and the exchange time
$T_{exc}^{CO_2/HCO_3^-}$ between solvated $CO_2$ and $HCO_3^-$ after recording
the spectra. $T_1$ was determined using saturation recovery with a
train of equispaced saturation pulses and logarithmically spaced
recovery times between 1 s and 128 s. For $T_2$ determination a

Carr–Purcell–Meiboom–Gill (CPMG) pulse sequence with an echo time of 5 ms was employed[57,58]. The chemical exchange between $CO_2$ and $HCO_3^-$ was assessed by 1D exchange spectroscopy (EXSY) using a selective inversion of the $HCO_3^-$ resonance by means of a Gauss shaped pulse with 100 Hz excitation bandwidth[59]. The chemical shifts were referenced to sodium trimethylsilylpropanesulfonate (DSS) (Sigma Aldrich, Munich, Germany) by means of an external measurement. The *in operando* experiments did not include DSS as the salt can interfere with the electrochemical measurements.

The $^{23}Na$ experiments were conducted in a similar manner, where, first, spectra were recorded for a total of 12 h with a time resolution of 6 min. Afterwards, $T_1$ and $T_2$ were determined using saturation recovery and CPMG experiments, respectively. $T_1$ was determined using saturation recovery with a train of equispaced saturation pulses and logarithmically spaced recovery times between 0.01 s and 0.96 s. For $T_2$ determination a CPMG pulse sequence with an echo time of 0.4 ms was employed[57,58]. The chemical shift was not referenced to any standard as the $^{23}Na$ experiments were used for validation purposes only.

**NMR data evaluation**. For post processing, a 1 Hz line broadening was applied to all $^{13}C$ spectra, and no line broadening was applied to the $^{23}Na$ spectra. NMR signals were deconvoluted by peak fitting using LMFIT version 0.9.14 for Python[60], employing a non-linear least-squares algorithm. The $^{13}C$ $HCO_3^-$ signal was fitted using two (OCV stage) or three (CA and CP stage) Lorentz peaks. For $^{23}Na$, one (OCV stage) or two (CA and CP stage) Lorentz peaks were fitted due to the limited resolution given by the inherently broad lines of the spin-3/2 resonances. As a note, the stability of the fit depended strongly on the number of free fit parameters and the overlap of the individual components. In case of a single spectrum the fit with either a large number of fit parameters and moderate overlap of the resonances, as for $^{13}C$ NMR, or a moderate number of parameters and a large overlap, as for $^{23}Na$ NMR, may lead to unstable fit results. However, due to the large number of spectra and a continuous time evolution of the signal components, the stability of the fit procedure was significantly improved by correlation of the individual fits within a time series.

The determination of relaxation time constants, and evaluation of the saturation recovery and CPMG experiments is depicted in the SI figs. S10–S13 and figs. S14–S17, respectively. For the evaluation of the EXSY experiments, $T_{exc}^{CO_2/HCO_3^-}$ was determined by fitting the evolution of the $CO_2$ signal integral $I(CO_2)$ as a function of the mixing time $\tau_m$ to

$$I(CO_2) = I_0(CO_2)\left\{1 - 2\left[\exp\left(-\frac{\tau_m}{T_{exc}^{CO_2/HCO_3^-} + T_1}\right) - \exp\left(-\frac{\tau_m}{T_1}\right)\right]\right\},$$

(14)

where $I_0$ is the signal integral at $\tau_m = 0$. This equation is valid under the conditions that the bicarbonate concentration exceeds the $CO_2$ concentration and that both species possess similar $T_1$ time constants[59]. The evaluation of the EXSY experiments is shown in more detail in the SI Fig. S18.

**DFT chemical shift calculations**. Density functional theory (DFT) based geometry optimisations and chemical shift calculations were conducted using ORCA version 5.0.2.[61] Geometry optimisation was done at B3LYP level with def2-TZVP[62] basis set, Grimme's D3 dispersion correction[63] and Resolution of Identity approximation. The solvent was modelled using a conductor-like polarisable continuum model (CPCM) for water and by explicitly including 5 water molecules each for the bicarbonate and the sodium ion. Chemical shift calculations were done with the same parameters with automatic generation of auxiliary basis sets[64]. Ab initio molecular dynamics (AIMD) simulations were done using the ORCA-MD module with identical functional and basis sets. Initial structures for the AIMD simulations were obtained by geometry optimising ion pair structures with specific distance constraints between Na and C atoms. The distance constraint was also applied throughout the AIMD trajectory. Velocities were initialised for a temperature of 298.15 K and maintained using a Nosé-Hoover thermostat with a time constant of 10 fs. Each MD trajectory was 1000 fs long with a time step of 1 fs. Chemical shift calculations were done for ion pair structures extracted from AIMD snapshots at intervals of 5 fs. Additional details of the chemical shift calculations are discussed in the supplementary methods in the SI.

## Data availability

NMR data are available for download at https://doi.org/10.26165/JUELICH-DATA/0GQJFC. Other data are available from the authors upon request.

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

## Acknowledgements

We would like to thank Philipp Schleker for the scientific discussions on solution and catalyst chemistry and Simone Köcher for her advice and suggestions on chemical shift calculations. We also gratefully acknowledge financial support by the German Federal Ministry of Education and Research (BMBF) within the Kopernikus Project P2X: Flexible use of renewable resources—research, validation and implementation of 'Power-to-X' concepts, and the Deutsche Forschungsgemeinschaft (DFG, German Research Foundation) under Germany's Excellence Strategy—Cluster of Excellence 2186 "The Fuel Science Center". Computational resources from RWTH Aachen University under project rwth1253 are acknowledged.

## Author contributions

Sven Jovanovic performed the *in operando* experiments and contributed major parts to the data analysis and evaluation. Davis Thomas Daniel performed the DFT chemical shift calculations. Peter Jakes, Steffen Merz, Rüdiger-A. Eichel, and Josef Granwehr contributed to the interpretation and evaluation of experiments and simulations, and supervised the research project. All authors have reviewed the manuscript.

## Funding

## Competing interests

The authors declare no competing interests
