## [Peer Review File · Communications Chemistry]

Reviewers' comments:

Reviewer #1 (Remarks to the Author):

In their manuscript titled: "In operando NMR investigations of the aqueous electrolyte chemistry during electrolytic CO₂ reduction" Jovanovic and coworkers describe a series of NMR spectroelectrochemistry experiments probing both ¹³C and ²³Na resonances. The NMR experiments appear to be expertly done, and the modeling seems to be chemically reasonable. The authors find that speciation and solvation are dependent on the applied potential. The provided data generally supports the conclusions; however, there are some concerns outlined below. Ultimately, the work is well done and the contents a valuable addition to the field. It is this reviewer's recommendation that the manuscript be reconsidered for publication in Communications Chemistry after the following revisions have been made.

Comments:

1. Numerous acronyms are used in the main text but are not defined until much later in the Methods section (i.e. OCV, CA, CP, etc.). Defining these in the main text will greatly enhance the readability of the manuscript.
2. The authors present a handful of NMR spectra, that focus on a few selected peaks; however, there is very little in terms of raw data. It would be helpful if the authors included a Supporting Information document with raw spectra/fit of inversion recovery experiment showing how T₁ values were calculated as well as a few full sweep NMR spectra during each phase of the electrochemical experiment (i.e. at OCV and during CA and CP).
3. In figure 4 the authors assign the upfield resonance to free ions and the downfield component to ion pairs based on electron density arguments. This is likely the case; however, chemical shift arguments for heteronuclear NMR may not always be so straightforward. It would be more rigorous if the authors could support this with literature examples or computations.

Reviewer #2 (Remarks to the Author):

In operando NMR investigations of the aqueous electrolyte chemistry during electrolytic CO₂ reduction

Comments

This is an excellent paper where operando ¹³C and ²³Na were employed to investigate the ion speciation and exchange during electrolytic reduction of CO₂ from near electrodes to bulk electrolyte solutions when an electric field is applied. The linkage of exchange between the different solvated species of bicarbonate in terms of free ion pairs, solvent associated ion pairs and contact ion pair is quite interesting. The experiments were done with care and interpretations are generally acceptable. Non-invasive, or non-destructive NMR is unique for studying the details of various kinds of ion pairs when in particular combined with T₁ and T₂ relaxation measurements. The extend of information obtained are impressive. I recommend the publication provided the following minor changes are made.

1. It is more difficult to read this manuscript at the beginning but then it becomes much easier to read. Please polish the introduction carefully so that the readers can have a good feeling immediately. You can achieve this by polishing the Language in some way.
2. Page 6/16: "The coalescence point is dependent on the B₀ field, as according to eq. (12) it is a function of the absolute peak separation in Hz, and not the relative one in ppm. Thus, under otherwise

identical conditions, a lower peak separation of the HCO_3^- signal is expected at lower field strength." When ppm is converted to Hz, for the same 10Hz a higher B_0 gives a reduced ppm. Therefore, the results in Figure 5 do not justify the minimal exchange rate.

Since T_1 and T_2 values are different between the different peaks at around 161 ppm in the absence of applied electric field, you can use this result as a strong argument that the peak splitting is not due to the B_0 inhomogeneity.

Reviewer #1 (Remarks to the Author):

In their manuscript titled: "In operando NMR investigations of the aqueous electrolyte chemistry during electrolytic CO₂ reduction" Jovanovic and coworkers describe a series of NMR spectroelectrochemistry experiments probing both ¹³C and ²³Na resonances. The NMR experiments appear to be expertly done, and the modeling seems to be chemically reasonable. The authors find that speciation and solvation are dependent on the applied potential. The provided data generally supports the conclusions; however, there are some concerns outlined below. Ultimately, the work is well done and the contents a valuable addition to the field. It is this reviewer's recommendation that the manuscript be reconsidered for publication in Communications Chemistry after the following revisions have been made.

The authors would like to thank the reviewer for the positive review of our work. As detailed below, we agree with your comments, which enabled us to considerably improve our manuscript.

Comments:

1. Numerous acronyms are used in the main text but are not defined until much later in the Methods section (i.e. OCV, CA, CP, etc.). Defining these in the main text will greatly enhance the readability of the manuscript.

Thank you for the remark. The sections of the manuscript were rearranged before submission, which is why the experimental section contains most of the definitions for the acronyms. In the revised manuscript, the full description of the acronyms was added at their first appearance in the text.

2. The authors present a handful of NMR spectra, that focus on a few selected peaks; however, there is very little in terms of raw data. It would be helpful if the authors included a Supporting Information document with raw spectra/fit of inversion recovery experiment showing how T₁ values were calculated as well as a few full sweep NMR spectra during each phase of the electrochemical experiment (i.e. at OCV and during CA and CP).

We agree that the original manuscript lacked raw data. In order to keep the main text as streamlined as possible, we decided to not add raw spectra or $T_1/T_2/T_{exc}$ evaluations in the manuscript itself, but we have added supporting information, which contain various NMR spectra with a considerably broader chemical shift range recorded during OCV, CA and CP for the ¹³C experiments at 14.1 T and 9.4 T as well as the ²³Na experiments at 14.1 T. In order to still be able to discern details in the spectra, we decided to show 13 spectra for each stage at a time resolution of 1 hour instead of all 120 spectra, which we recorded 6 minutes apart. In addition, we included all fit plots that were used to evaluate T_1 , T_2 and T_{exc} of the HCO₃⁻ components and the CO₂ signal (SI page 1 – 11). Finally, we referred to the SI in the main text (page 13 & 14).

3. In figure 4 the authors assign the upfield resonance to free ions and the downfield component to ion pairs based on electron density arguments. This is likely the case; however, chemical shift arguments for heteronuclear NMR may not always be so straightforward. It would be more rigorous if the authors could support this with literature examples or computations.

Thank you for your suggestion. We agree that this interpretation lacked theoretical validation. We have included DFT calculations of the chemical shift difference between solvated free ions and solvated ion pairs using two different hydration sphere models.

Overall, the theoretical results are in good agreement with the experimental results. A discussion of the DFT calculations can be found in the manuscript on page 6 and the SI in section 2. Davis Thomas Daniel, who performed the DFT calculations, has been added as a co-author.

Reviewer #2 (Remarks to the Author):

In operando NMR investigations of the aqueous electrolyte chemistry during electrolytic CO₂ reduction

Comments

This is an excellent paper where operando ¹³C and ²³Na were employed to investigate the ion speciation and exchange during electrolytic reduction of CO₂ from near electrodes to bulk electrolyte solutions when an electric field is applied. The linkage of exchange between the different solvated species of bicarbonate in terms of free ion pairs, solvent associated ion pairs and contact ion pair is quite interesting. The experiments were done with care and interpretations are generally acceptable. Non-invasive, or non-destructive NMR is unique for studying the details of various kinds of ion pairs when in particular combined with T1 and T2 relaxation measurements. The extend of information obtained are impressive. I recommend the publication provided the following minor changes are made.

The authors would like to thank the reviewer for the highly positive reception of our manuscript. As detailed below, we have improved the manuscript based on the reviewer's helpful comments and remarks.

1. It is more difficult to read this manuscript at the beginning but then it becomes much easier to read. Please polish the introduction carefully so that the readers can have a good feeling immediately. You can achieve this by polishing the Language in some way.

We improved the language of the introduction throughout. We also included sentences or passages as transition between certain sections in order to improve readability.

2. Page 6/16: "The coalescence point is dependent on the B₀ field, as according to eq. (12) it is a function of the absolute peak separation in Hz, and not the relative one in ppm. Thus, under otherwise identical conditions, a lower peak separation of the HCO₃⁻ signal is expected at lower field strength." When ppm is converted to Hz, for the same 10Hz a higher B₀ gives a reduced ppm. Therefore, the results in Figure 5 do not justify the minimal exchange rate.

Frist, writing this comment we noticed an error in the formula eq. 12, as well as an inconsistency. Correctly, eq. 12 should equal the exchange rate at the coalescence point. In addition, ΔΩ is the frequency difference in rad/s. As we give the peak separation in Hz, we substituted Ω by $\nu = \frac{\Omega}{2\pi}$. This results in:

$$k_{exc}^{colesc} = \frac{\pi\Delta\nu}{\sqrt{2}}$$

To be precise, this equation is only strictly correct for two exchanging species of equal concentration, but as ion pairs and free ion signal integrals are on the same

order of magnitude, eq.12 is still a good approximation, The corresponding calculations were done correctly and are not affected, just the written formula was incorrect. We adjusted eq. 12 in the manuscript and apologize for the inconvenience.

Second, we improved our estimation of the exchange rate between free ions and xSIPs by calculating exchange spectra based of the modified Bloch equations of Gutowsky and Holm. Using these equations, exchange spectra at 9.4 T could be calculated based on the experimental data at 14.1 T, and then compared to the observed exchange spectra at 9.4 T. Based on these considerations, an exchange rate of 15 1/s could be estimated, which is in line with our previous considerations. Nonetheless, we adapted the manuscript and considered the improved estimation method on pages 3 and 7.

Since T1 and T2 values are different between the different peaks at around 161 ppm in the absence of applied electric field, you can use this result as a strong argument that the peak splitting is not due to the B0 inhomogeneity.

Thank you for the comment. We are, however, hesitant to state whether the differences in relaxation time constants can exclude that the appearance of the shoulder (i.e. the green component) is NOT due to a B0 inhomogeneity. Instead, we theorize in our previous publication that the connected electrodes, even in absence of an electric field, may lead to new relaxation pathways in proximity of the electrodes. For example, conduction electrodes in the metal are paramagnetic, and thus solution in the very proximity of the electrode surface may be affected by paramagnetic relaxation. Thus, the observed shoulder at OCV may actually be a combined result of these relaxation effect AND B_0 inhomogeneities.

REVIEWERS' COMMENTS:

Reviewer #1 (Remarks to the Author):

The authors have adequately addressed this reviewer's comments and the manuscript is now suitable for publication.